# Pattern of Residual Submucosal Involvement after Neoadjuvant Therapy for Rectal Cancer: A Rationale for the Utility of Endoscopic Submucosal Resection

**DOI:** 10.3390/medicina59101807

**Published:** 2023-10-11

**Authors:** Haidy Elazzamy, Monika Bhatt, Paul Mazzara, Mohammed Barawi, Amer Zeni, Amr Aref

**Affiliations:** 1Pathology Department, Ascension St. John Hospital, Detroit, MI 48236, USApmazzara@yahoo.org (P.M.); 2Gastroenterology, Ascension St. John Hospital, Detroit, MI 48236, USA; 3Rectal Surgery, Ascension St. John Hospital, Detroit, MI 48236, USA; amerzeni@gmail.com; 4Radiation Oncology, Ascension St. John Hospital, Detroit, MI 48236, USA

**Keywords:** submucosal, neoadjuvant therapy, rectal cancer, endoscopic resection

## Abstract

*Background and Objectives*: Full-thickness trans anal local excision for tumors with favorable response following neoadjuvant therapy for locally advanced rectal cancer (LARC) is a common strategy for organ preservation, but it could be associated with a high rate of postoperative complications. We describe the incidence and pattern of submucosal involvement in surgical specimens following neoadjuvant therapy for LARC and whether limiting local excision of the residual tumor bed to only mucosal/submucosal layers of the rectal wall is sufficient for accurately predicting the ypT status of residual cancer, providing a pathological rationale to replace full-thickness local excision by endoscopic submucosal resection. *Materials and Methods*: This was a single-institution retrospective study conducted at a teaching community hospital. We reviewed clinical and pathological findings with slides of 82 patients diagnosed with LARC treated at our center between 2006 and 2020. Eligibility criteria mirrored our current organ preservation trials. *Results*: No tumor was found in surgical specimens in 28 cases (34%). Additionally, 4, 22, 27, and 1 cases were staged as ypT1, ypT2, ypT3, and ypT4, respectively. Residual malignant cells were found in the submucosal layer in 98% of cases with ypT+ stage, with ‘skip lesions’ in only 2% of cases. *Conclusions*: A very high incidence of submucosal involvement is noticed in residual tumors after neoadjuvant therapy, providing pathological rationale to study the role of endoscopic submucosal resection as a restaging tool for tumors with favorable response after neoadjuvant therapy when organ preservation strategy is pursued. This study was limited by its retrospective design and relatively small number of patients.

## 1. Introduction

Preoperative therapy with concurrent chemoradiotherapy or total neoadjuvant therapy (TNT) followed by Total Mesorectal Excision (TME) is recognized as the standard of care for the management of locally advanced rectal cancer [1]. Although this approach yields an excellent local control rate, it can be associated with frequent and severe long-term side effects that adversely impact the patients’ quality of life (Low Anterior Resection Syndrome) [2,3,4,5]. Therefore, many centers are investigating the use of an “organ preservation strategy” as an alternative treatment for selected tumors with favorable responses to neoadjuvant therapy. The goal of this approach is to replace the routine use of TME with either watchful waiting (WW) or full-thickness local excision (LE) when a favorable response to preoperative therapy is observed.

When the WW strategy is adopted after the initial complete tumor response to preoperative treatment is detected, no surgery is offered unless tumor regrowth is noticed during the surveillance period. The successful implementation of WW assumes that the tumor regrowth can be detected soon after it becomes clinically apparent and salvage TME can be performed shortly after. These requirements are important to avoid the increased risk of distant metastasis development and/or ultimate pelvic failure following salvage TME. Patients’ compliance with close follow-up and improved methods of tumor restaging are essential to meet these requirements [6]. We are not aware of any reported prospective study that compared the safety of the WW approach with the standard of care treatment using TME.

When LE is performed and the subsequent histological examination confirms the ypT0-1 status, the expected local control rate is approximately 95%. Completion TME is recommended when the histological examination reveals a more extensive residual disease than ypT0-1-R0 and the surgical eradication of the residual cancer is accomplished within a few weeks following LE, thus avoiding the risk of undetected malignancy for a prolonged period which may increase the risk of distant metastasis. One prospective trial and several retrospective reports confirmed the safety of the organ preservation approach with LE compared to the standard treatment with TME [7,8,9,10,11]. Despite the clear benefits of the LE approach, many centers have abandoned it in favor of the WW method because of concerns about unacceptable postoperative morbidity [12,13,14,15]. We previously reported that LE, when performed after neoadjuvant therapy for the sole purpose of ensuring a complete microscopic response, can be limited to the removal of the entire rectal wall just adjacent to a residual mucosal abnormality without the need for the excision of any margin of the surrounding normal tissue, a procedure we named “limited local excision” [16,17,18].

We aim, in the current work, to investigate the possibility of further refinement of the limited local excision procedure by limiting the surgical excision to only the mucosal and submucosal layers of the rectal wall underneath the residual mucosal abnormality without resection of the muscle layer or the perirectal fat tissue. We launched this investigation to determine whether the presence or absence of the residual tumor cells in the submucosal layer of the rectal wall correlates accurately with the ypT status and the involvement of the muscle and perirectal fat tissues by cancer cells. This information is essential for future investigation of the potential role of endoscopic submucosal resection as a restaging tool after a favorable response to preoperative treatment, as will be discussed later in this manuscript.

## 2. Materials and Methods

This study was approved by the Ascension St. John Institutional Review Board, IRB Study ID RMIJ20210408, 9 August 2023.

### 2.1. Patient Selection

Patients aged >18 years who were clinically staged with cT2-3 or cN0-1 invasive adenocarcinoma of the rectum between 2006 and 2020 were included. Patients were eligible for this study only if their clinical parameters mirrored the current eligibility criteria for our ongoing organ preservation protocols. Patients who would not have been offered organ preservation according to our current treatment policy, such as those with cT1N0, cN + ve disease with a known lymph node size of >1 cm, or cT4, and patients presenting with distant metastatic spread, were excluded.

### 2.2. Treatment

Each patient received preoperative chemoradiation therapy with or without induction chemotherapy prior to definitive surgery, which included lower anterior resection, abdominal perineal resection, or local excision.

The time of days between the end of radiation treatment and definitive surgery was calculated.

### 2.3. Slide Selection and Scoring System

Original slides from each patient’s definitive surgery were evaluated to determine the presence of residual adenocarcinomas. Gross pathology specimens were processed and sectioned according to the guidelines of the College of American Pathologists at the time of surgery.

Microscopic slides representative of the center of the residual abnormality were carefully reviewed. Tumor regression grade according to Ryan’s methodology [19], pathological stages, and mucosal, submucosal, muscle, and fat involvement with residual adenocarcinoma were noted. The incidence and pattern of submucosal involvement of the residual tumor were carefully examined and tabulated. We noticed different patterns of submucosal involvement and developed a classification system, Mazzara–Elazzamy classification, to categorize this variation. We classified submucosal involvement as focal or Type I if it involved one 10× field (1 mm) or less in its largest dimension. Submucosal involvement with at least two tumor foci separated by at least a low-power field (4×) was classified as patchy or Type II. Finally, any tumor greater than a 10× field, regardless of the number of foci, was described as diffuse or Type III. Table 1 depicts the *Mazzara–Elazzamy classification*. We also quantified the residual disease in the submucosa in millimeters(Figure 1) We attempted to correlate the incidence of skip lesions with different patterns of submucosal involvement.

For all patients included in the study, original hematoxylin and eosin-stained slides of definitive surgical specimens were collected and reviewed initially by two authors (HE, MB). A second review of specimens was performed by an experienced colorectal pathologist (PM) who determined the final description of each slide examination.

## 3. Results

### 3.1. Patient Demographics and Clinical Characteristics

In total, 82 patients (45 men and 37 women) were included in this study. The median age of the patients was 64 years (36–87 years). The median tumor distance from the anal verge was 7 cm (0–15 cm). The majority of patients presented with cT3 tumors and only two patients had cT2 disease. In 50 cases, no clinically significant lymphadenopathy was detected, whereas 32 cases were staged as cN + ve.

### 3.2. Treatment Details

Neoadjuvant treatment consisted of concurrent chemoradiotherapy with five fluorouracil/capecitabine. Induction FOLFOX chemotherapy was administered in later years in 14 cases (17%). Radiation therapy doses ranged between 45 and 50.4 Gy, with only two patients receiving a radiation dose of 54 Gy. The time interval between the completion of chemoradiotherapy and surgical resection ranged from 28 to 244 days with a median value of 53 days. Seventy-two patients had undergone anterior resection while five patients were treated by abdominoperineal resection. Full-thickness local excision was performed in five patients. Table 2 summarizes the clinical characteristics of patients.

### 3.3. Pathological Outcomes

In total, 28 cases (34%) were staged as ypT0, while 4, 22, 27, and 1 were staged as ypT1, ypT2, ypT3, and ypT4, respectively. Mucosal involvement was observed in 27 patients (50% of patients with residual cancer). The submucosa was involved in 53 of the 54 (98%) patients with ypT+ disease. Table 3 describes the pathological outcome after neoadjuvant therapy, as well as the *Mazzara–Elazzamy classification* pattern of submucosal involvement and the amount of residual submucosal disease associated with each type of involvement.

## 4. Discussion

There have only been a few descriptions of residual microscopic disease in locally advanced rectal cancer after neoadjuvant therapy. Meterissian et al. [20] retrospectively reviewed the treatment of patients at MD Anderson Hospital between 1990 and 1992. All 37 patients considered were clinically staged as T3 via endoscopic ultrasound and received preoperative concurrent 5FU and external radiation therapy. Tumor responses to treatment were assessed six weeks after the completion of radiation therapy. A pathological complete response was achieved in 30% of the patients. Mucosal involvement of residual disease was detected in 65% of patients with residual disease in the rectal wall and in 25% of the entire cohort. The incidence of submucosal involvement has not yet been reported.

Duldulao et al. [21] conducted a retrospective review of pathological slides of patients enrolled in a prospective multi-institutional phase II trial designed to determine the optimal timing of surgery for advanced rectal cancer after chemoradiotherapy [22]. The study cohort included patients with clinical stage II and III adenocarcinoma of the rectum. All patients underwent TME following a waiting period after the completion of chemoradiotherapy, the duration of which ranged between 6 and 16 weeks. One-third of the patients considered did not receive consolidation multi-agent systemic chemotherapy (waiting time of 6 weeks), and the other two-thirds received either two or four cycles of consolidation multi-agent systemic chemotherapy with a waiting interval between completion of chemoradiotherapy and TME of 11–16 weeks. The available surgical specimen tumor blocks after TME were retrieved, recut, and analyzed to determine the exact location of any residual cancer cells within the different layers of the rectal wall. A complete pathological response was observed in 26% of specimens examined. The mucosa was involved in only 15% of specimens with residual invasive cancer after neoadjuvant therapy, while the submucosa was involved in 52% of cases.

A similar retrospective pathological study by Xiao et al. [23] involved a group of patients with more advanced tumors (57% of patients presented with T4 disease). Accordingly, the patients received more aggressive neoadjuvant therapy, with all patients treated with concurrent radiotherapy and multi-agent chemotherapy. Additionally, all patients underwent TME 6–8 weeks after completing radiotherapy. Post-surgical tumor blocks were retrieved, recut, and prepared for analysis. A complete pathological response was observed in 28% of cases, and the incidence of mucosal involvement was only 14%. Submucosal involvement occurred in 27% of cases. Microscopic disease was detected in the muscle layer and perirectal fat of 47% and 46% of specimens, respectively.

Although there is some treatment variation in our cohort with only 17% receiving induction chemotherapy, we do not think that this heterogeneity could affect the result of our main outcome measurement as nearly all specimens demonstrated submucosal involvement with residual cancer. For the same reason (the high incidence of submucosal involvement), there is no influence of the clinical staging or the length of time intervals between completion of radiation and surgery on the incidence of submucosal involvement in ypT+ tumors. In addition, we did not expect to detect a correlation between the pattern of submucosal involvement using the *Mazzara–Elazzamy* classification and the ypT or ypN status again because of the presence of submucosal involvement in nearly all cases with residual cancer. Nevertheless, we propose this classification system as a possible initial framework for further investigations by other institutions in this space.

Our results are quite different from those of Duludao and Xiao. The reason for this discrepancy is unclear; however, it may be related to differences between patient cohorts. In particular, our cohort tended to present with less advanced disease than those of prior studies. Our high pCR rate of 34%, despite the use of systemic chemotherapy in only 17% of cases, may reflect the early stage of disease among patients of our study. In addition, we reviewed the exact pathologic slides that were used during the initial post-surgical evaluation, especially those slides taken from the center of residual abnormalities since this region carries the maximum residual tumor burden. Further, we did not have to resort to re-cutting the tumor blocks, as was performed in the studies by Duludao and Xiao. It is important to note that 25% of cases of submucosal involvement were by *Mazzara–Elazzamy classification* Type I. This focal involvement could possibly have been missed if enough care was not taken to examine pathology slides obtained from the center of the residual mucosal abnormality, the region of the expected largest bulk of residual tumors.

The difference between our results and those of Duludao and Xiao can have potentially significant clinical implications. We detected submucosal involvement in 98% of patients with residual microscopic malignancy. This finding clearly needs to be confirmed by future studies with larger data sets, and if confirmed, it could imply that the incidence of submucosal “skip lesions” in >ypT1 cases is extremely low among our and similar cohorts. This observation provides a rationale for future investigation of the role of the endoscopic submucosal resection technique to determine the presence or absence of residual microscopic malignancy after completion of neoadjuvant treatments with a great degree of certainty and may limit the need for FTLE in many clinical scenarios. This new approach may decrease postoperative morbidity values and may also improve post-treatment rectal function. Indeed, if postoperative morbidity following ESR is significantly reduced, particularly with regard to avoiding complicating subsequent completion TME (if required), the approach may function as a much-needed restaging tool that reduces the likelihood of erroneously performing TME on lesions presumed to harbor viable malignancy before subsequent examination of an amputated rectum reveals pCR. The incidence of this condition in contemporary series is approximately 10%. Another potential advantage of ESR is its applicability early throughout WW, particularly in cases of a near-complete clinical response to minimize the incidence of tumor regrowth, which can be as high as 40% in some series [24]. Our results may also be useful in assessing cases in which traditional local excision is performed with the intent of resecting the entire thickness of the rectal wall, but for technical reasons, only the mucosa and submucosa with or without part of the muscle layer-, are removed.

A clear drawback of the use of ESR is its inability to distinguish between stage yp1 and higher stages if a residual tumor is detected in the submucosal layer. In this scenario, some patients with ypT1 disease may undergo unnecessary completion TME. This shortcoming will be important only for centers that offer FTLE for ypT1 lesions and not for centers that recommend completion TME for all patients with residual disease after neoadjuvant therapy, regardless of the ypT stage.

Whether ESR can be performed successfully after preoperative radiotherapy remains to be seen. However, very early reports indicate the feasibility of this approach [25,26,27]. Our group is currently planning to initiate a pilot study to determine the feasibility and safety of the procedure.

A strength of our study is the fact that the decision was made a priori to assess our research question, which has direct clinical applications. Further, the study had well-defined eligibility criteria. It is the only report to determine the exact incidence of submucosal skip lesions within the rectal wall. Studies by Duludao and Xiao were not designed to answer this specific question, although a possible range of the incidence of submucosal skip lesions of 41–44% can be estimated from their submitted data. Notably, the occurrence of skip lesions estimated from previous studies was much higher than that which was reported here. Our study was limited by the small number of cases considered and its retrospective design. Our findings require confirmation using larger datasets.

## 5. Conclusions

This pathological study determined that the residual microscopic malignant cells, following neoadjuvant therapy for locally advanced rectal cancer, are rarely present in the muscle layer of the rectal wall or the perirectal fat without being detected in the more superficial submucosal layer. The incidence of “skip lesions” in our study was only 2%. Our results provide a pathological basis to investigate the potential role of endoscopic submucosal resection to restage tumors following the completion of neoadjuvant therapy for locally advanced rectal cancer when organ preservation strategy is contemplated. This pathological study suggests that limiting the local excision to only the mucosa and submucosa layers of the rectal wall directly beneath the residual mucosal abnormality, without the need for removing the muscle layer or perirectal fat, may be sufficient for predicting the presence or absence of residual rectal cancer following neoadjuvant chemoradiotherapy. Our results provide a basis to investigate the role of endoscopic submucosal resection to restage tumors following the completion of neoadjuvant therapy for locally advanced rectal cancer when the organ preservation strategy is contemplated. Our findings need to be confirmed using larger datasets.

## Figures and Tables

**Figure 1 medicina-59-01807-f001:**
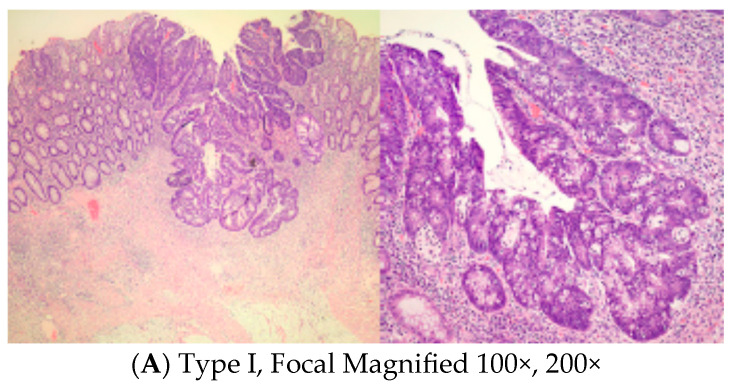
Low and high power photographs for Mazzara–Elazzamy classification.

**Table 1 medicina-59-01807-t001:** Mazzara–Elazzamy classification of submucosal involvement.

Pattern	Definition
TYPE I	Focal: residual tumor involving one 10× field or less
TYPE II	Patchy: residual tumor involving two distinct areas with at least one low power field (4×) of separation
TYPE III	Diffuse: residual tumor involving an area greater than a 10× field regardless of the number of foci present

**Table 2 medicina-59-01807-t002:** Patient Parameters.

Characteristic	Value	
Gender (#)	Male	45
	Female	37
Age (Y)	Median and range	64 (36–87)
Distance from anus (cm)	Median and range	7 (0–15)
Clinical T stage (#)	T2	2
	T3	80
Clinical N stage (#)	N0	50
	N+	32
Interval between completion of radiation and surgery (D)	Median and range	53 (30–244)

# Number; Y Years; D Days.

**Table 3 medicina-59-01807-t003:** Pathological Outcomes.

Category	Value/Quantity	
* TRG	TRG I	29
	TRG II	34
	TRG III	20
** YPT	YPT0	28
	YPT1	3
	YPT2	22
	YPT3	28
	YPT4	1
*** YPN	N0	49
	N1 N2	31 1
	Nx	1
***Mazzara–Elazzamy classification* of submucosal involvement**		
**Type**	**Quantity**	**Median Size/Range in MM**
Type I	14	0.3 (0.1–0.5)
Type II	6	0.7 (0.3–1.2)
Type III	33	0.8 (0.3–2.8)

* Tumor Regulation Grade; ** Post-Treatment Pathological Tumor Staging; *** Post-Treatment Pathological Nodal Staging.

## Data Availability

Data will be available upon reasonable request to the authors.

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
