# Peer review of "Pattern of Residual Submucosal Involvement after Neoadjuvant Therapy for Rectal Cancer: A Rationale for the Utility of Endoscopic Submucosal Resection"

_medicina, 2023, doi:10.3390/medicina59101807_

Round 1

Reviewer 1 Report

Thank you for inviting me to review this manuscript. I have read it with great interest. The manuscript offers valuable insights into the pattern of submucosal involvement following neoadjuvant therapy for locally advanced rectal cancer. The focus on endoscopic submucosal resection (ESR) as a restaging tool is novel and has potential clinical implications. The discussion surrounding the feasibility, advantages, and challenges of ESR is particularly noteworthy.

Suggestions for Improvement:

Introduction

Some sentences are quite dense, and breaking them down into shorter sentences can improve readability.

The introduction could benefit from smoother transitions between the different treatment approaches (WW, FTLE) and the rationale behind each.

Explicit Aims: While you've mentioned the aim of the study, it might be helpful to state it more explicitly at the end of the introduction.

Methods

Consider breaking down this section into subsections to make it easier for readers to follow different aspects of the methodology (e.g., Patient Selection, Slide Examination, Scoring System) for structural Organization.

While you've described the Mazzara-Elazzamy Classification, you could explicitly mention how this classification system was developed, how the criteria for focal, patchy, and diffuse submucosal involvement were determined, and why these classifications are relevant.

Mention how inter-rater agreement was established among the reviewers, especially when using a classification system like Mazzara-Elazzamy. Include information about inter-rater reliability or intra-rater agreement in slide evaluation.

Results

Please add explanations for the abbreviations in Table 3.

Consider organizing the information in a logical sequence by first discussing patient demographics, clinical characteristics, and then moving on to treatment details and pathological outcomes.

The manuscript presents a well-conducted study with valuable insights into submucosal involvement after neoadjuvant therapy for rectal cancer. The investigation of endoscopic submucosal resection's potential utility is particularly noteworthy. With some minor revisions for clarity, organization, and elaboration, the manuscript has the potential to contribute meaningfully to the field of rectal cancer treatment strategies.

Minor editing of English language required

Author Response

  • Introduction section has been rewritten with more clarification of the aim of the study. 
  • The method section has been broken down and subsections were inserted. 
  • Explanation of the Mazzara-Elazzamy classification development was inserted. 
  • This study was not designed to determine the inter-observer agreement between different pathologists. Therefore we cannot comment on that point. 
  • Explanation for the abbreviations in Table 3 was inserted. 
  • The information in the results section was organized and we used subsections to describe the information. 

Reviewer 2 Report

In this paper, the authors reviewed clinical and pathological findings with slides of 82 patients diagnosed with LARC, to determine whether the presence or absence of tumor cells in the submucosal layer correlates accurately with the ypT status of the surgical specimens. They showed that residual malignant cells were found in the submucosal layer in 98% of cases with ypT+ stage, with ‘skip lesions’ in only 2% of cases. The study was supported by some data, but some issues may need the authors to be explained.

1.       It should have scale bars in Fig.1 and use high resolution images.

2.       The introduction is not attractive enough for readers, and the authors should give more clear knowledge about why they perform this study.

3.       The conclusion is not clear.

The English language can be improved, and there are few errors

Author Response

  • The introduction section has been rewritten
  • The conclusion section has been rewritten
  • The original slides have been sent back to the warehouse and it is not possible to retrieve them in a timely fashion (the publisher asked us to submit our response to the reviewers and the revised manuscript within 3 business days)